# NEURAL ARCHITECTURE SEARCH FOR TINYML WITH REINFORCEMENT LEARNING

## ABSTRACT

Deploying Deep Neural Networks (DNNs) on microcontrollers (TinyML) is a common trend to process the increasing amount of sensor data generated at the edge, but in practice, resource and latency constraints make it difficult to find optimal DNN candidates. Neural Architecture Search (NAS) is an excellent approach to automate this search and can easily be combined with DNN compression techniques commonly used in TinyML. However, many NAS techniques are not only computationally expensive, especially hyperparameter optimization (HPO), but also often focus on optimizing only a single objective, e.g., maximizing accuracy, without considering additional objectives such as memory consumption or computational complexity of a model, which are key to making deployment at the edge feasible. In this paper we propose a novel NAS strategy for TinyML based on multi-objective Bayesian optimization (MOBOpt) and an ensemble of competing parametric policies trained using Augmented Random Search (ARS) Reinforcement Learning (RL) agents. Our methodology aims at efficiently finding tradeoffs between a DNN's predictive accuracy, memory consumption on a given target system, and computational complexity. Our experiments show that we outperform existing MOBOpt approaches consistently on different data sets and architectures such as ResNet-18 and MobileNetV3.

## 1 INTRODUCTION

The deployment of DNNs on embedded devices is restricted to the constraints imposed by the target microcontroller platform. Constraints such as <1MB Flash, <512Kb RAM, and clock speeds in the low MHz range make the design of DNN models for such platforms challenging. In general, several conflicting objectives and constraints such as memory availability, inference time, and power consumption of the deployed DNN model must be considered.

Despite extensive research on Neural Architecture Search (NAS), there is still no definitive method that is both fast and robust. Many black-box optimization approaches (Zoph & Le, 2017; Real et al., 2019; White et al., 2021) work reliably, but are inefficient. Differentiable NAS strategies such as DARTS (Liu et al., 2019) have robustness problems, as it can be observed that the architectures found often do not generalize well because the search overfits the validation data set (Zela et al., 2020). Finally, zero-cost NAS strategies that have been proposed to allow rapid network specialization for different target platforms and resource constraints, such as Once-for-all (Cai et al., 2020) or PreNAS Wang et al., 2023, while fast, do not provide accurate information about an architecture's performance, but only statistics, such as the number of FLOPs (White et al., 2022).

In this work, we investigate how to improve the efficiency of NAS for DNN deployment on microcontrollers using a combination of Multi-objective Bayesian optimization (MOBOpt), deep compression, and Augmented Random Search (ARS) Mania et al. (2018). To this end, we formulate the architecture search as a hyperparameter optimization (HPO) problem (Bergstra et al., 2013), where the parameters influence the complexity of a given architecture using filter pruning (Li et al., 2017) and weight quantization (Gholami et al., 2021). This allows us to optimize DNNs, considering the objectives of top-1 accuracy, memory consumption, and FLOPS.

As our main contribution, we propose a novel Bayesian-optimization based solver using an ensemble of competing locally parameterizable policies that are iteratively trained on the underlying surrogate model using an ARS RL agent. Our experiments show that this RL-based strategy helps to improve

the effectiveness of our NAS approach in finding optimal DNNs for microcontroller deployment compared to other state-of-the-art multi-objective optimization strategies for multiple use cases while introducing negligible additional overhead for training the RL agents. The architectures proposed by our approach can be deployed directly on common microcontrollers without further (re-)training.

The rest of the paper is structured as follows. Sec. 2 discusses related work. Sec. 3 provides background on multi-objective Bayesian optimization. Sec. 4 describes our RL agent and how we integrate it into our MOBOpt DNN hyperparameter exploration process. We evaluate our approach in Sec. 5 on several public datasets with different types of neural networks (Sec. 5.2) and on a synthetic optimization problem (Sec. 5.4). Sec. 6 concludes.

## 2   RELATED WORK

Different approaches for efficient design and deployment of DNNs on embedded platforms have been prominently discussed in scientific research: Dedicated architectures such as the MobileNet class (Howard et al., 2017; Sandler et al., 2018; Howard et al., 2019) have been proposed, introducing scaling parameters and specialized layers, i.e., depthwise-separable convolutions, to control size and inference time. Deep Compression features a number of techniques, including pruning (Li et al., 2017), which dynamically removes weights during training, and weight quantization (Jacob et al., 2018), which reduces the resolution at which weights are represented. Knowledge distillation (KD) (Hinton et al., 2015) has also been used for DNN compression (Ashok et al., 2018; Cao et al., 2018). However, DNN optimization using KD remains challenging as the student often fails to match the predictions of the teacher (Stanton et al., 2021; Cho & Hariharan, 2019). Based on DNN compression deployment pipelines for embedded devices have been proposed: Han et al. (2016) combine network pruning, weight quantization and Huffman coding and follow-up work compares different pruning and quantization methods (Deutel et al., 2023). Other well known DNN deployment pipelines are MCUNet (Lin et al., 2020; 2021) and Tensorflow Lite Micro (David et al., 2021).

Although NAS is an intensively researched topic, harnessing its power to find DNN deployment candidates for microcontroller platforms is still a largely unsolved problem. Previous approaches (Wang et al., 2023; Cai et al., 2020; 2018) mainly focus on zero-cost NAS that derive architecture variants for a given target platform and a given set of resource constraints from a well-trained "supernet" structure or focus on hardware aware search for larger mobile platforms (Tan et al., 2019; Dong et al., 2018). In contrast, although largely unexplored in the context of NAS for microcontroller targets, Multi-Objective Bayesian Optimization (MOBOpt) is one of the most widely used black-box optimization methods in NAS for solving HPO problems. (Elsken et al., 2018; White et al., 2021; Chitty-Venkata & Somani, 2022; White et al., 2023). As training DNNs is time- and resource-intense the reduced sample complexity of Bayesian optimization is beneficial compared to otherwise popular genetic optimization (Lu et al., 2019; Deb et al., 2002). When optimizing continuous hyperparameters, algorithms using Gaussian processes (GP) are most prominent (Knowles, 2006). An extension of this optimization approach that specifically aims to tackle problems with high-dimensional search spaces suggests the use of locally modeled confidence regions (Eriksson et al., 2019; Daulton et al., 2022).

The use of RL for MOOpt has been extensively studied: The general approach is to consider the search space as the action space of the RL agent, whose reward is the performance of the trained architecture on unseen data. Approaches to optimization using RL differ in how they represent the policies they use, e.g., He et al. (2018) use Deep Deterministic Policy Gradient (DDPG) agents, Moffaert & Nowé (2014) use tabular Q-learning, Kim et al. (2022) use single-step RL approaches, and Li et al. (2020) utilize actor-critic algorithms.

In this work, we combine the sampling efficiency of GPs with the expressive power of RL agents. Compared to previous work on RL for MOOpt, our approach does not directly apply RL to the search space of the MOOpt problem, but instead uses RL to improve the optimization performed on the GPs. To address the problem of expending large amounts of resources to train RL agents, we instead propose to train a set of competing policies within MOBOpt that implement candidate sampling using very simple Augmented Random Search (ARS) (Mania et al., 2018) RL agents. Hence, our approach allows for efficient exploration of DNN architectures directly deployable on microcontroller.

## 3 BACKGROUND

The goal of multi-objective optimization (MOOpt) is to either maximize or minimize a set of objectives. Assume, without loss of generality, maximizing some set of objective functions $f(x) = [f_1(x), \ldots, f_n(x)] \in \mathbb{R}^n$, where $n \geq 2$, while satisfying a set set of constraints $g(x) \geq 0 \in \mathbb{R}^V$ where $V \geq 0$, $x \in \mathcal{X} \subset \mathbb{R}^d$, and $\mathcal{X}$ is a compact set. Usually, there exists no single solution $x^*$ that maximizes all objectives while also satisfying all $V$ constraints.

**Definition 3.1.** An objective function evaluation $f(x)$ *Pareto-dominates* $f(x')$, denoted as $f(x) \succ f(x')$, if $f_m(x) \geq f_m(x')$ for all $m = 1, \ldots, M$ and there exists at least one $m \in \{1, \ldots, M\}$ such that $f_m(x) > f_m(x')$. (Daulton et al., 2022)

**Definition 3.2.** A set of Pareto-optimal tradeoffs $P(x)$ over a set of samples $X \subseteq \mathcal{X}$ is called the *Pareto front* (PF). $\mathcal{P}(X) = \{f(x) : x \in X, \nexists x' \in X \text{ s.t. } f(x') \succ f(x)\}$. The *feasible PF* is defined as $\mathcal{P}_{feas}(X) = \mathcal{P}(\{x \in X : g(x) \geq 0\})$. (Daulton et al., 2022)

Hence, the goal of MOOpt is to identify an approximate PF $\mathcal{P}(X_n)$ of the true PF $\mathcal{P}(X)$ within a search budget of $n$ evaluations. In cases where the true PF is not known, the quality of $\mathcal{P}(X_n)$ is usually assessed using the Hypervolume (HV) indicator, also called the S-metric.

**Definition 3.3.** The *Hypervolume indicator* $HV(\mathcal{P}(X)|r)$ is the $M$-dimensional Lebesgue measure $\lambda_M$ of the region dominated by $\mathcal{P}(X)$ and bounded from below by a reference point $r \in \mathbb{R}^M$. (Daulton et al., 2022)

A feasible reference point is commonly derived from domain knowledge of the problem. The HV also allows to compare multiple PFs as long as they were calculated with the same reference point and share the same objective values.

A way of improving the sample efficiency of multi-objective optimization is by using Bayesian Optimization (BO) aiming at minimizing the number of required evaluations of the given parameter space. BO treats the objective functions as black-boxes and places a prior over them, thereby capturing beliefs about their behaviour. Over time, as new samples are collected, the prior is updated iteratively to form the posterior distribution over the objective functions, in our case by using GPs. The posterior, in turn, is used in each iteration to determine the next sample-point to evaluate by employing an acquisition function as a heuristic to quantize the "usefulness" of the sample. Traditionally, the acquisition function is also responsible for balancing the exploration and exploitation tradeoff. Furthermore, since we are considering multiple objectives, we scale and accumulate the objective values to form a single-objective problem (Knowles, 2006; Paria et al., 2020).

A commonly used acquisition function is the Expected Improvement (EI) Močkus (1975)

$$EI(x) = \mathbb{E} \max(f(x) - f^*, 0) \tag{1}$$

where $f^*$ is the objective value of the best observed sample so far. For a given Bayesian model the EI can be evaluated with an integral over the posterior distribution either analytically or by Monte-Carlo sampling Wilson et al. (2017), which allows to approximate the EI as

$$qEI(x) \approx \frac{1}{N} \sum_{i=1}^{N} \max_{j \ldots q} \{\max(\xi_i - f^*, 0)\}, \xi_i \sim \mathbb{P}(f(x) \mid \mathcal{D}), \tag{2}$$

where $q$ is the number of samples considered jointly and $\mathbb{P}(f(x) \mid \mathcal{D})$ is the posterior distribution of the function $f$ at $x$ given the data $\mathcal{D}$ observed so far.

## 4 METHOD

In this work we solve a multi-objective HPO problem using Bayesian optimization and RL. Different from other Bayesian optimization algorithms, our methodology iteratively finds new sets of hyperparameters using an ensemble of competing multi-layer perceptron (MLP)-based RL agents, in particular ARS (Mania et al., 2018), to optimize the acquisition function. Our policies only use two linear layers with a hidden layer size of 64 resulting in a couple of thousand trainable parameters each. ARS is an RL agent which learns a set of linear (or MLP) policies $\pi_\theta$, parameterized by a set of vectors $\theta$, one for each layer, for controlling a dynamic environment $\mathbb{E}_\xi$, with $\xi$ encoding

---

**Algorithm 1** Multi-Objective Bayesian Optimization with Augmented Random Search

---

**Parameters:** search budged $J$, learning rate $\alpha$, directions sampled per iteration $N$, rollout horizon $H$, constant exploration noise $v$, number of top-performing directions to use $k$, number of objectives $n$, number of parameters $p$, number of ARS agents $L$, objective function $f(x) = [f_1(x), \ldots, f_n(x)]$

Create initial prior by evaluating $f(x)$ several times using Latin-Hypercube sampling
**for** $i \leftarrow 1$ **to** $J$ **do**
    Fit GPs by maximising the marginal likelihood of previous evaluations of $f(x)$.
    Select $\{x_1, x_2, \ldots, x_L\}$ initial states from $\mathcal{P}_{feas}$ of previous evaluations using k-means clustering.
    **repeat**
        **for** $l \leftarrow 0$ **to** $L$ **do**
            Sample directions $\varphi_1, \varphi_2, \ldots, \varphi_N \in \mathbb{R}^{p \times n}$ with i.i.d. standard normal entries.
            Collect the summed reward of the MC-sampled and scaled posteriors, see Equations 7 and 8, of $2N$ rollouts over the horizon $H$ from the GP using local policy $\pi_l$ and initial state $x_l$.

$$\pi_{l,j,k,+}(x) = (\theta_{l,j} + v\varphi_k)x \tag{3}$$
$$\pi_{l,j,k,-}(x) = (\theta_{l,j} - v\varphi_k)x \tag{4}$$
$$\pi_{l,j}(x) = \theta_{l,j}x \tag{5}$$

            for $k \in \{1, 2, \ldots, N\}$.
            Sort the directions $k$ by $\max\{r(\pi_{l,j,k,+}), r(\pi_{l,j,k,-})\}$, denote by $\varphi_k$ the $k$-th largest direction, and by $r(\pi_{l,j,k,+})$ and $r(\pi_{l,j,k,-})$, the corresponding rewards.
            Update the parameters $\theta_l$ of policy $\pi_l$ using the top-$k$ performing rollouts (with $\theta_{l,0} = 0$)

$$\theta_{l,j+1} = \theta_{l,j} + \frac{\alpha}{b\sigma_R} \sum_{k=1}^{b} \left( r(\pi_{l,j,k,+}) - r(\pi_{l,j,k,-}) \right) \varphi_k \tag{6}$$

            where $\sigma_R$ is the standard deviation of the rewards used in the update step.
        **end for**
    **until** ARS termination condition is satisfied
    Perform rollouts for all initial states $x_l \in P_{feas}$ using policy $\pi_l$. For each $\pi_l$ select the rollout yielding the highest reward and evaluate using $f(x)$ and extend the prior with the new evaluation
**end for**

---

the randomness of the environment, maximizing/minimizing an average reward/loss $r(\pi_\theta, \xi)$. ARS achieves this by optimizing over the set of parameters $\theta$ by utilizing derivative-free optimization with noisy function evaluations and for each layer iteratively performing updates using directions of the form $\frac{r(\pi_{\theta+\nu\varphi}, \xi_1) - r(\pi_{\theta-\nu\varphi}, \xi_2)}{\nu}$ for two i.i.d. random variables $\xi_1$ and $\xi_2$, $\nu$ a positive real number (0.008 in our case), and $\varphi$ a zero mean Gaussian vector with i.i.d. standard normal entries. The two rewards $r(\pi_{\theta+\nu\varphi}, \xi_1)$ and $r(\pi_{\theta-\nu\varphi}, \xi_2)$ are obtained by collecting two sets of trajectory rollouts from the system of interest.

We initialize the training of our competing ARS agents with different samples from the current feasible Pareto front $\mathcal{P}_{feas}(X)$. To find the most diverse set of samples, we perform k-means clustering on the Pareto front if it consists of more elements than we can train policies. To query the objective values, i.e., accuracy, memory consumption, and FLOPS, for a proposed set of hyperparameters of a given DNN architecture, our objective function evaluator first trains, prunes, and quantizes the DNN. Afterwards, the DNN is converted into source code using an automatic code generator to accurately assess memory usage of weights and activation tensors. Our implementation performs this procedure using the compression and deployment pipeline proposed by Deutel et al. (2023), who describe a configurable pipeline that supports both DNN pruning and full weight quantization. Following their results, we focus our experiments on iterative filter pruning and post-training static quantization.

We integrated ARS into MOBOpt as a parameterized, trainable solver, which can thus be seen as an alternative to traditional Bayesian solvers, see Algorithm 1. The derivative-free random sampling

approach implemented by ARS is computationally inexpensive and highly parallelizable, allowing for efficient GPU-accelerated execution.

We define the states $x_0, x_1, \ldots, x_L$ of our $L$ competing agents as vectors of the parameters of the search space of our optimization problem. Each agent performs actions based on its trainable policy $\pi_\theta$ by altering the parameter values of its state vector within given bounds. We train the competing policies of our ARS agents for each sample after evaluating the objective function by performing rollouts on posteriors sampled from the GP. We fit them by maximizing the marginal likelihood of the previously queried evaluation of the objective function (see also Williams & Rasmussen (1995)). The policy training performed for each sample ends after a maximum number of training steps has been conducted. To obtain the initial prior for the GP, we propose candidates during the first couple of iterations of the optimization using Latin Hypercube sampling (McKay et al., 2000). We chose this approach over random sampling because we found that by using Latin Hypercube sampling, we could significantly improve the expressiveness of our priors and therefore enhance the performance of all evaluated GP-based sampling strategies.

As a reward $r(x)$, we first scale our $n = 4$ objectives (accuracy, ROM/RAM memory consumption, FLOPS) to a single objective value using augmented Chebyshev scalarization (Knowles, 2006), see Eq. (7), where $\rho = 0.005$ is a small positive value and $\lambda$ is a weight vector that we draw uniformly at random at the beginning of each sample. We then calculate the difference between the scaled objective and the best previously visited (scaled) objective $f^*$ as the reward, see Eq. (8).

$$f_\lambda(x) = \max_{j=1}^{n}(\lambda_j f_j(x)) + \rho \sum_{j=1}^{n} \lambda_j f_j(x) \tag{7}$$

$$r(x) = \begin{cases} f_\lambda(x) - f_\lambda(f^*), & \text{if } (f_\lambda(x) - f_\lambda(f^*)) > 0 \\ 1e^{-3}(f_\lambda(x) - f_\lambda(f^*)), & \text{otherwise} \end{cases} \tag{8}$$

To avoid having to compute the expected value over the posterior of the GP when computing the acquisition function, a common approach is to use Monte-Carlo (MC) sampling as an approximation to evaluating the integral over the posterior distribution. We implement a quasi-MC sampling approach to estimate the reward used to train the ARS agents with a fixed set of base samples.

As a result, the reward $r(x)$ used by our ARS agents is comparable to acquisition functions used in traditional Bayesian optimization. In particular, our approach is similar to the expected improvement (EI) used by ParEgo (Knowles, 2006), with the notable difference that we consider penalized "negative improvements" instead of clamping them to zero. The main reason for clamping "negative improvement" in combination with a Monte Carlo-based evaluation of the GPs is to increase the attractiveness of regions with high uncertainty, and is therefore the mechanism of EI to control the tradeoff between exploration and exploitation. However, we argue that we can still achieve the desired tradeoff if we instead penalize "negative improvements" by multiplication with a small constant factor, while allowing our ARS policies to learn from situations where the current objective function evaluation is worse than the best previously visited one.

After training the competing ARS agents, we perform several rollouts for each of them starting from their respective starting points selected from $\mathcal{P}_{feas}(X)$. We then select the policy that yielded the overall best reward to propose the next set of hyperparameters to be evaluated on the objective function in the next sample. A variant of this approach would be to consider a set of best-performing parametrizations as candidates for a batched optimization in different segments of the Pareto front.

## 5 EVALUATION

We applied our multi-objective DNN hyperparameter optimization on two use-cases with 150 samples each: (1) image classification (32x32 RGB images from CIFAR10 (Krizhevsky, 2012)) using a reduced version of ResNet18 (He et al., 2016) (three residual blocks instead of four, 1.6M initial parameters) and (2) time-series classification of (daily) human activities (DaLiAc) (Leutheuser et al., 2013) with a window length of 1024 datapoints, using a down-scaled version of MobileNetv3 Howard et al. (2019) (2.8M initial parameters). Compared to models commonly used by relevant benchmarks for embedded AI, e.g., TinyML Perf (Banbury et al., 2021), we did not reduce the initial numbers parameters as our optimization process uses pruning and therefore dynamically adjusts the exact number of neurons during training. We trained our models using Stochastic Gradient Decent (SGD)

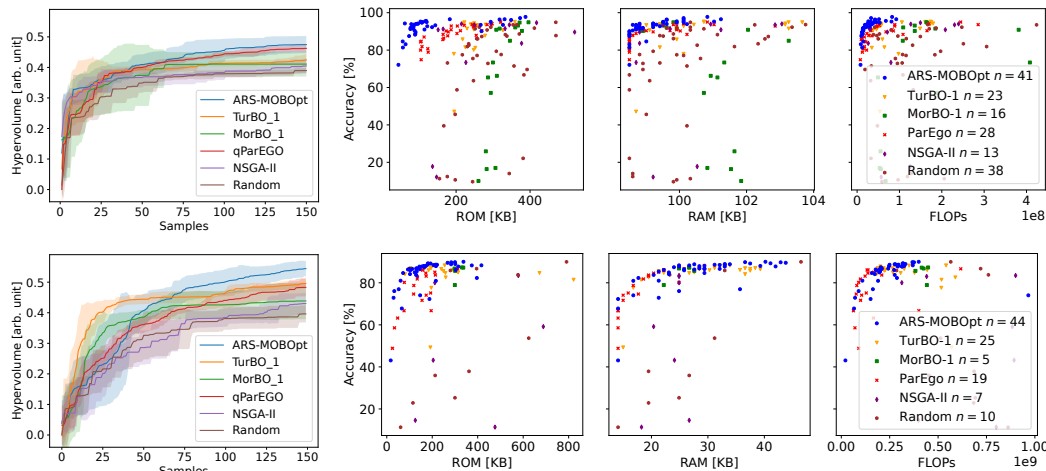

Figure 1: ARS-MOBOpt compared to baseline approaches (top row: MobileNetv3 on DaLiAc; bottom row: ResNet on CIFAR10). Left column: Our approach (ARS-MOBOpt) outperformed all other optimization. Remaining columns: feasible Pareto sets proposed by our RL-based MOBOpt of DNNs (blue) compared to other Bayesian and evolutionary approaches (in both cases we find samples dominating those proposed by the other algorithms while meeting imposed target constraints).

with an exponential learning rate decay. For the optimization, we chose 256 KB of RAM, 1 MB of ROM and $1e^9$ FLOPS as constraints imposed by common microcontroller platforms.

The detailed list of hyperparameters considered by our optimizer and their sampling intervals derived from expert knowledge can be found in Appendix 7.1. We consider seven parameters controlling general properties of DNN training (training length, batch size, properties of the optimizer and learning rate) and four that control the filter pruning. We use the automated gradual pruning algorithm (Zhu & Gupta, 2017) to generate our pruning schedule. It takes a start and end epoch calculated relatively to the number of training epochs and a number of iterative steps as an input. Our optimization considers the pruning sparsity hyperparameter, i.e., how many filters are removed, for each convolutional layer separately. Therefore, the search space considered increases with the depth of the optimized DNN as for each prunable layer an additional pruning sparsity parameter is added which means that the search space is extended by one dimension.

## 5.1 ALGORITHMIC BENCHMARK

As our first experiment we compared the quality of our approach (ARS-MOBOpt) to several well-known Bayesian optimization strategies (ParEGO (Knowles, 2006), TurBO (Eriksson et al., 2019), MorBO (Daulton et al., 2022)), one evolutionary approach (NSGA-II (Deb et al., 2002)), and a random sampling approach as a baseline, see Fig. 1. All Bayesian optimization strategies were implemented using BoTorch (Balandat et al., 2020). We gave each optimization strategy a search budget of 150 samples to evaluate the objective functions and monitored the improvement of the feasible hypervolume over the samples. For all tested Bayesian strategies the first 10 samples were conducted as priors using Latin Hypercube sampling (McKay et al., 2000). We normalized our objectives accuracy, ROM and RAM consumption, and FLOPs to their respective constraints and used the maximum feasible objective value as a common reference when computing the hypervolume. We performed 5 independent seeds for each strategy and provide the observed variance as well.

Fig. 1 shows the results for two common DNN architectures, MobileNetv3 and ResNet18, which we trained using two datasets, DaLiAc and CIFAR10. The plots show the observed hypervolume over the number of optimization samples. Therefore, the resulting curves, one for each optimization strategy, describe the improvement of the hypervolume. First, all informed solvers outperformed the independent random sampling baseline on both use-cases. Second, all the approaches that are based on Bayesian optimization, i.e., TurBO, MorBO, ParEGO and ARS-MPBOpt, which consider the constraints of a very tight search budget of 150 samples, performed better then the evolutionary sampling strategy (NSGA-II). Additionally, for both use-cases a significant Hypervolume improve-

Table 1: Excerpt from the results of the multivariate datasets of the UEA & UCR time series classification benchmark (Bagnall et al., 2017) for our ARS strategy (columns 3–7) and ParEGO (columns 8–12). We show the achieved hypervolume after 150 trials as well as the maximum accuracy. We also show the minimum memory and FLOPs required to achieve at least 70% accuracy.

| Dataset | InceptionTime Acc. [%] | ARS | | | | | ParEGO | | | | |
|---|---|---|---|---|---|---|---|---|---|---|---|
| | | HV | Acc. [%] | ROM [Kb] | RAM [Kb] | FLOPs | HV | Acc. [%] | ROM [Kb] | RAM [Kb] | FLOPs |
| Epilepsy | 98.55 | **0.97** | 97.81 | $1.21 \times 10^5$ | $4.92 \times 10^3$ | $3.57 \times 10^6$ | 0.93 | 94.16 | $6.70 \times 10^4$ | $3.39 \times 10^3$ | $2.05 \times 10^6$ |
| BasicMotions | 100.00 | 0.96 | 97.50 | $1.31 \times 10^5$ | $3.00 \times 10^3$ | $2.75 \times 10^6$ | **0.97** | 97.50 | $8.67 \times 10^4$ | $2.50 \times 10^3$ | $1.49 \times 10^6$ |
| Cricket | 98.61 | **0.96** | 97.22 | $1.00 \times 10^5$ | $2.99 \times 10^4$ | $1.60 \times 10^7$ | 0.95 | 95.83 | $7.61 \times 10^4$ | $3.05 \times 10^4$ | $1.01 \times 10^7$ |
| ERing | 87.78 | **0.96** | 96.67 | $1.00 \times 10^5$ | $1.99 \times 10^3$ | $2.03 \times 10^6$ | 0.93 | 93.33 | $7.32 \times 10^4$ | $1.23 \times 10^3$ | $1.00 \times 10^6$ |
| ArticularyWordRecognition | 98.33 | **0.94** | 95.64 | $1.51 \times 10^5$ | $6.18 \times 10^3$ | $4.19 \times 10^6$ | 0.92 | 92.36 | $8.07 \times 10^4$ | $5.33 \times 10^3$ | $2.13 \times 10^6$ |
| UWaveGestureLibrary | 87.81 | **0.91** | 91.67 | $1.10 \times 10^5$ | $6.94 \times 10^3$ | $5.78 \times 10^6$ | **0.91** | 91.67 | $7.21 \times 10^4$ | $4.09 \times 10^3$ | $3.03 \times 10^6$ |
| NATOPS | 96.11 | **0.86** | 87.22 | $1.29 \times 10^5$ | $5.78 \times 10^3$ | $2.79 \times 10^6$ | 0.78 | 78.33 | $1.00 \times 10^5$ | $4.94 \times 10^3$ | $1.12 \times 10^6$ |
| SelfRegulationSCP1 | 83.96 | **0.79** | 79.85 | $1.05 \times 10^5$ | $2.24 \times 10^4$ | $1.78 \times 10^7$ | 0.78 | 78.73 | $7.81 \times 10^4$ | $2.24 \times 10^4$ | $7.15 \times 10^6$ |
| PEMS-SF | – | 0.76 | 81.50 | $1.38 \times 10^5$ | $5.55 \times 10^5$ | $7.98 \times 10^6$ | **0.79** | 84.97 | $2.38 \times 10^5$ | $5.56 \times 10^5$ | $5.72 \times 10^7$ |
| Heartbeat | 58.05 | **0.73** | 74.51 | $8.16 \times 10^4$ | $9.92 \times 10^4$ | $8.14 \times 10^6$ | 0.73 | 74.02 | $6.25 \times 10^4$ | $9.92 \times 10^4$ | $2.79 \times 10^6$ |

ment can be observed when using our ARS-based sampling strategy with 3000 sampling directions, a top-$k$ selection percentage of 1%, and a rollout horizon of 4 to train the competing policies. This implies that by using a trainable RL-based strategy to determine new promising parametrizations as the optimization progresses, our algorithm makes much better use of the knowledge about the search space encoded in the surrogate model than the other non-trainable Bayesian sampling approaches.

In addition, we provide qualitative results for both considered use cases, see the remaining columns in Fig. 1. The plots show the Pareto optimal samples for our approach (in blue) compared to all other algorithms tested on MobileNetv3/DaLiAc (top row) and Resnet/CIFAR10 (bottom row). For both use cases, ARS-MOBOpt was able to find samples that are Pareto-dominant over those proposed by the other algorithms, while satisfying all imposed target constraints ($<$1MB ROM, $<$256KB RAM, $<$1e9 FLOPs). Moreover, for both use cases, the number of elements $n$ in the two sets of feasible Pareto optimal configurations resulting from our approach are larger than the sets found by the other approaches, implying that our algorithm was able to provide a larger variety of Pareto optimal deployable DNNs.

To get more insight into the general performance of ARS, we tested it on a subset of the multivariate datasets of the UEA & UCR time series classification benchmark (Bagnall et al., 2017) and compared ourselves to ParEGO using the same setup as described above, see Table 1 for all data sets on which reasonable models with a predictive accuracy of at least 70% are found (and Table 3 in Appendix 7.2 for a complete listing). To provide context, we also list reference results for InceptionTime (Ismail Fawaz et al., 2020) that we extracted from Bagnall et al. (2017). We let both ARS and ParEGO optimize a CNN architecture with a single regular convolution followed by 10 depth-wise separable convolutions with batchnorm and with a total of 5 squeeze and excitation blocks (Hu et al., 2018) after convolutions 10, 8, 6, 4, and 2, followed by adaptive average pooling and a single linear layer for classification (2.6m initial parameters). Our approach achieves a better hypervolume than ParEGO in 7 of 10 data sets (and is en par on one data set). If we also consider data sets where we found models with predictive accuracies below 70% our approach outperforms ParEGO in approx. two-thirds of the data sets, see Appendix 7.2. From the results we also see that compared to ParEGO, our algorithm was generally better at finding DNNs with higher accuracy scores, but often at the cost of higher memory and FLOPs requirements.

## 5.2 ROBUSTNESS AND HYPERPARAMETER SELECTION

Our proposed ARS sampling strategy can be configured with a number of hyperparameters that affect the behavior of the ARS algorithm, see Alg. 1. The quality of the results obtained by our algorithm can vary depending on the parameterization, see Fig. 2: In general, choosing a learning rate of $1e^{-3}$, an exploration noise of $1e^{-2}$, a top-$k$ selection rate of 1%, a number of sampling directions around 3000, and a rollout horizon of 4 yielded the best results. In the following, we want to further discuss the impact of key hyperparameters, i.e., the number of sampling directions $N$, the top-$k$ percentage of directions selected for policy updates, and the number of steps $H$ taken for each episode (rollout horizon), to provide a broader understanding of their effects.

First, especially for complex optimization problems with a high-dimensional parameter space, a high number of sampling directions significantly improved the hypervolume that our ARS-based sampling strategy was able to achieve, compared to a low number of directions, which often resulted

in a below-average performance, see Fig. 2a. However, we also found, that it is not feasible to increase the number of directions indefinitely to support more and more complex problems, as this also significantly increases the time required to train the ARS policies and the memory consumption of our algorithm. We found that the exact number of sampling directions which is required to obtain a competitive result is highly correlated with the complexity of the underlying optimization problem.

Second, when increasing the top-$k$ percentage of sampling directions selected by ARS to perform policy updates, we observed that only selecting a small subset of $1\%$ of the sampled directions to perform policy updates improved the hypervolume achieved by ARS compared to using larger subsets, see Fig. 2b. Our observation is thus consistent with Mania et al. (2018)'s argument for introducing the hyperparameter as part of ARS. The authors point out that including all observed rewards in the parameter update can be detrimental to performance because outliers can easily introduce bias.

Third, we found that an optimal episode length (rollout horizon) is at $H = 4$ steps, see Fig. 2c. Both a significantly smaller and larger episode length on average resulted in a decreased performance. Still, we want to point out that we observed a significantly larger variance for longer episode lengths compared to shorter episode lengths. This is consistent with the increased variance in the estimation of total expected return in MC-like algorithms like ARS (Sutton & Barto, 2018).

## 5.3 COMPARISON TO MCUNET AND ZERO-SHOT NAS

We compare the performance of our optimization framework with MCUNet (Lin et al., 2020; 2021) and ProxylessNAS (Cai et al., 2018) for the CIFAR10 task, see Appendix 7.3 for a full list of results. Our main findings are that our NAS approach was able to find network candidates with slightly higher accuracy than the MCUNet variants we tested, while offering a better memory footprint but higher computational complexity.

We also report results for a configuration of MCUNet that we optimized for CIFAR10 using ProxylessNAS, a zero-shot NAS framework. As a result, it does not produce a Pareto front of trained DNNs, but instead emits a DNN configuration that needs to be trained separately after optimization. For CIFAR10, ProxylessNAS produced a configuration estimated at $86.53\%$ and $27.16M$ MACs, which after training turned out to be slightly overestimated ($82.33\%$ and $25.5M$ actual) and not competitive to some of the architectures found by our framework, see Fig. 1.

## 5.4 INTUITION

To better understand why our RL-based solver can improve on other Bayesian solvers like ParEgo, we analyzed both using a synthetic optimization problem. Suppose we minimize a two-dimensional single objective optimization problem (Passino, 2005) using both ParEgo and our approach, see Fig. 3. We chose this problem because it presents an interesting challenge: it has a landscape with large areas with little to no gradient combined with some steep minima and maxima. We provide the topography of the objective function in Appendix 7.4 and show the respective Bayesian surrogates of ParEgo and ARS-MOBOpt as well as the topographies of their expected improvement (EI) after 40 trials in Fig. 3a and 3b. For our approach (ARS-MOBOpt), we also show the rollouts performed during the validation of the local policies for five selected start points (red crosses) after the last (40th) sample.

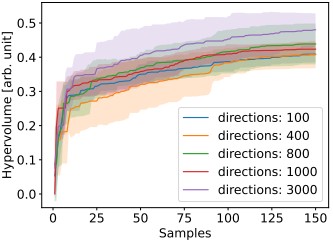
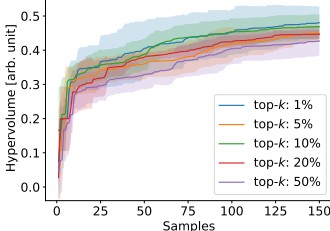
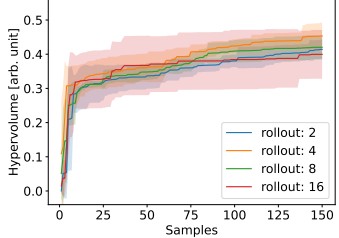

(a) Number of *sampling directions* $N$ performed for each step.

(b) Percentage of *top-k* performing samples selected for policy updates.

(c) *Rollout Horizon H* performed by the competing policies.

Figure 2: Detailed results examining the effects of the three key ARS hyperparameters on the achieved hypervolume, 5 seeds each. MobileNetv3, 1.6M init. params, DaLiAc dataset, window length 1024.

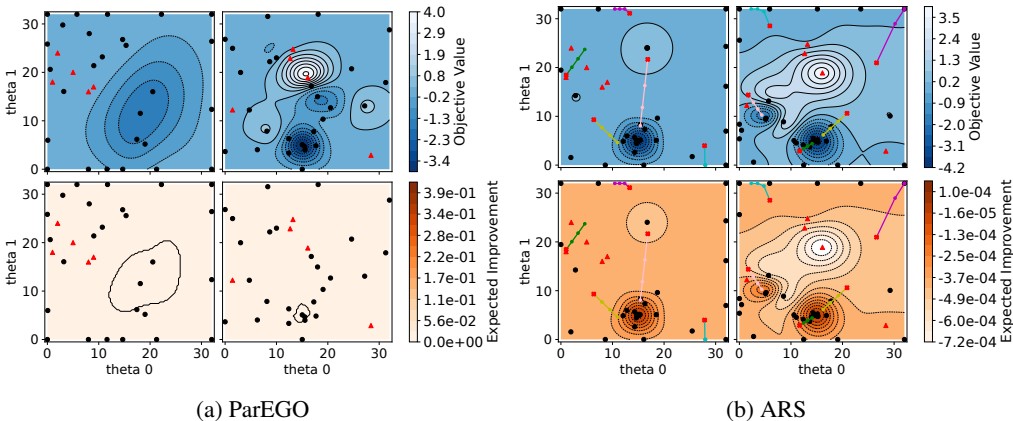

(a) ParEGO             (b) ARS

Figure 3: Topography of Bayesian surrogates (top row) and EI (bottom row) after 40 samples for both ParEGO and ARS (ours) given two sets of priors marked with red triangles on a synthetic minimization problem (Passino, 2005) (ground truth see Appendix 7.4, global minimum at $\theta_0 = 15$, $\theta_1 = 5$). For ARS, the rollouts of the trained policies of the 40th sample are shown as lines with their starting points marked by red crosses.

We manually selected two sets of priors (marked as red triangles) to provide both algorithms with interesting starting situations: First, a situation where all samples are close together in a region with almost no gradient, see top left in Fig. 3a and 3b. This initialization is interesting because it provides almost no information about the environment, requiring extensive exploration by the algorithms before they can exploit found minima. Second, a more general starting situation with a broader distribution of prior samples, see top right in Fig. 3a and 3b. This initialization focuses more on evaluating how well the algorithms can apply the exploration-exploitation tradeoff and how fast they can find minima. For both priors, we took care not to include samples close to the global minimum.

Looking at the results proposed by ParEgo (marked as black dots) after 40 samples in Fig. 3a, it is clear that for both sets of priors the algorithm seems to have a clear focus on exploration, leading to a wide sampling of the search space. However, looking at the two examples, this can lead to situations where even though the global optimum was either not found (left column in Fig. 3a) or it was not fully exploited (right column in Fig. 3a). Furthermore, we argue that this focus on exploration is mainly a result of the definition of the EI as it does not consider any "negative improvement". Looking at the topography of the EI for both examples in the last row of Fig. 3a, it is clear that this can result in large parts of the acquisition function having no gradient. We argue that this is not optimal when relying on gradient-based solvers, as the ParEgo implementation typically does.

In comparison, if we look at the result of our proposed ARS-based approach in Fig. 3b, it can be seen that our competing local policies, visualized as lines with red crosses as their starting point, have become experts in solving the environment around their starting point and can therefore provide sound solutions even in regions without significant gradients. Therefore, compared to ParEgo, we observed less undirected exploration and a clearer focus on exploiting environmental knowledge, compared to Fig. 3a. However, despite this more pronounced focus on exploitation, due to the competing ensemble of agents, our solver was still able to consistently escape local minima, e.g. see Fig. 3b (right col.).

## 6 CONCLUSION

We presented a NAS methodology for efficient multi-objective DNN optimization for microcontrollers based on Bayesian optimization and Augmented Random Search (ARS). Since the objectives accuracy, ROM consumption, RAM consumption and FLOPs were expensive to evaluate and we were faced with a limited search budged, we focused on performing time-efficient optimization of both DNN hyper- and compression-parameters to enable an optimal deployment on microcontroller platforms. We provided results for two different problems considering two datasets and DNN architectures and showed empirically that our algorithm was able to yield a better feasible Pareto front compared to well-known Bayesian optimization strategies like ParEGO, TurBO or MorBO.

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

# 7 Supplementary Material

## 7.1 Training Details & Optimization Variables

We present a multi-objective Bayesian optimization approach for DNN hyperparameter optimization that uses deep compression, i.e., pruning and quantization, to find feasible tradeoffs between DNN accuracy and resource consumption. Furthermore, we present a novel solver based on Augmented Random Search. An illustration of our process can be found in Fig. 4, with its implementation based on PyTorch, the Optuna (Akiba et al., 2019) framework and BoTorch (Balandat et al., 2020). The implementation consists of two building blocks: (1) the multi-objective optimization loop marked in blue and (2) a set of objective function evaluators marked in orange, for which we use the implementation discussed in Deutel et al. (2023). The optimization loop iteratively proposes new sets of hyperparameter configurations using ARS as its solver until a termination condition is met.

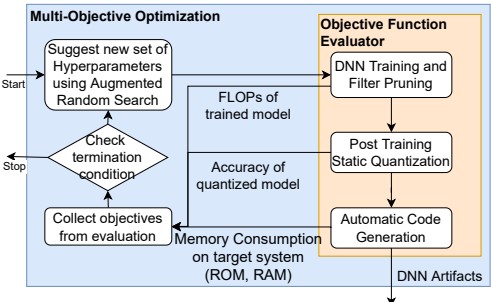

Figure 4: Overview of our proposed RL-based multi-objective Bayesian DNN hyperparameter optimization approach for microcontroler targets.

We provide a complete listing of the hyperparameters that are optimized and their search space intervals in Table 2. The first seven parameters are hyperparameters related to general DNN training, while the last five are related to compression. Note that each trainable layer (convolutions and linear layers) has its own optimizable pruning sparsity, and that we use iterative pruning with the pruning start and pruning end parameters defined relative to the number of training epochs. The search space intervals are chosen either based on expert knowledge, e.g., learning rate, momentum or weight decay, or based on time and hardware constraints of the optimiztion, e.g., training epochs and batch size.

Table 2: Hyperparameters considered by the multi-objective optimization. Learning rate schedule, pruning start, and end are relative to the epochs.

| Parameter | Search Space | Interval |
|---|---|---|
| epochs | uniform | $[100, 500]$ |
| batch size | uniform | $[20, 200]$ |
| learning rate (lr) | log. uniform | $[1e^{-5}, 1e^{-2}]$ |
| momentum | log. uniform | $[0.7, 0.99]$ |
| lr schedule | uniform | $[0.4, 0.9]$ |
| lr gamma | uniform | $[0.4, 0.9]$ |
| weight decay | log. uniform | $[0.6, 0.99]$ |
| pruning start | uniform | $[0.0, 0.6]$ |
| pruning end | uniform | $[0.8, 0.95]$ |
| pruning steps | uniform | $[1, 20]$ |
| pruning sparsity | uniform | $[0.1, 0.99]$ |

In Table 3 we provide all results we obtained for the multivariate datasets of the UEA & UCR time series classification benchmark (Bagnall et al., 2017). We provide results for our ARS strategy (columns 3–7) and ParEGO (columns 8–12). For both strategies, we show the achieved hypervolume after 150 trials as well as the maximum accuracy. We also show the minimum memory and FLOPs required to achieve at least 70% accuracy. If no trained networks were found that met these requirements (marked in gray), we show the memory and FLOPs consumption of the network that achieved the highest accuracy. We provide single objective reference results for InceptionTime (Ismail Fawaz et al., 2020) as taken from Bagnall et al. (2017).

Table 3: Results for the multivariate datasets of the UEA & UCR time series classification benchmark (Bagnall et al., 2017) for our ARS strategy (columns 3–7), ParEGO (columns 8–12) and InceptionTime (column 1) as a reference.

| Dataset | InceptionTime Acc. [%] | ARS HV | Acc. [%] | ROM [Kb] | RAM [Kb] | FLOPs | ParEGO HV | Acc. [%] | ROM [Kb] | RAM [Kb] | FLOPs |
|---|---|---|---|---|---|---|---|---|---|---|---|
| Epilepsy | 98.55 | **0.97** | 97.81 | $1.21 \times 10^5$ | $4.92 \times 10^3$ | $3.57 \times 10^6$ | 0.93 | 94.16 | $6.70 \times 10^4$ | $3.39 \times 10^3$ | $2.05 \times 10^6$ |
| BasicMotions | 100.00 | 0.96 | 97.50 | $1.31 \times 10^5$ | $3.00 \times 10^3$ | $2.75 \times 10^6$ | **0.97** | 97.50 | $8.67 \times 10^4$ | $2.50 \times 10^3$ | $1.49 \times 10^6$ |
| Cricket | 98.61 | **0.96** | 97.22 | $1.00 \times 10^5$ | $2.99 \times 10^4$ | $1.60 \times 10^7$ | 0.95 | 95.83 | $7.61 \times 10^4$ | $3.05 \times 10^4$ | $1.01 \times 10^7$ |
| ERing | 87.78 | **0.96** | 96.67 | $1.00 \times 10^5$ | $1.99 \times 10^3$ | $2.03 \times 10^6$ | 0.93 | 93.33 | $7.32 \times 10^4$ | $1.23 \times 10^3$ | $1.00 \times 10^6$ |
| ArticularyWordRecognition | 98.33 | **0.94** | 95.64 | $1.51 \times 10^5$ | $6.18 \times 10^3$ | $4.19 \times 10^6$ | 0.92 | 92.36 | $8.07 \times 10^4$ | $5.33 \times 10^3$ | $2.13 \times 10^6$ |
| UWaveGestureLibrary | 87.81 | 0.91 | 91.67 | $1.10 \times 10^5$ | $6.94 \times 10^3$ | $5.78 \times 10^6$ | **0.91** | 91.67 | $7.21 \times 10^4$ | $4.09 \times 10^3$ | $3.03 \times 10^6$ |
| NATOPS | 96.11 | **0.86** | 87.22 | $1.29 \times 10^5$ | $5.78 \times 10^3$ | $2.79 \times 10^6$ | 0.78 | 78.33 | $1.00 \times 10^5$ | $4.94 \times 10^3$ | $1.12 \times 10^6$ |
| SelfRegulationSCP1 | 83.96 | **0.79** | 79.85 | $1.05 \times 10^5$ | $2.24 \times 10^4$ | $1.78 \times 10^6$ | 0.78 | 78.73 | $7.81 \times 10^4$ | $2.24 \times 10^4$ | $7.15 \times 10^6$ |
| PEMS-SF | – | 0.76 | 81.50 | $1.38 \times 10^5$ | $5.55 \times 10^5$ | $7.98 \times 10^6$ | **0.79** | 84.97 | $2.38 \times 10^5$ | $5.56 \times 10^5$ | $5.72 \times 10^7$ |
| Heartbeat | 58.05 | **0.73** | 74.51 | $8.16 \times 10^4$ | $9.92 \times 10^4$ | $8.14 \times 10^6$ | 0.73 | 74.02 | $6.25 \times 10^4$ | $9.92 \times 10^4$ | $2.79 \times 10^6$ |
| HandMovementDirection | 36.49 | **0.60** | 60.81 | $1.71 \times 10^5$ | $2.36 \times 10^4$ | $1.01 \times 10^7$ | 0.45 | 45.95 | $9.95 \times 10^4$ | $1.86 \times 10^4$ | $9.72 \times 10^6$ |
| FingerMovements | 56.00 | **0.59** | 60.00 | $1.62 \times 10^5$ | $6.27 \times 10^3$ | $3.10 \times 10^6$ | 0.58 | 59.00 | $1.68 \times 10^5$ | $6.97 \times 10^3$ | $4.34 \times 10^6$ |
| SelfRegulationSCP2 | 47.22 | **0.58** | 58.33 | $1.05 \times 10^5$ | $5.87 \times 10^4$ | $3.92 \times 10^7$ | 0.56 | 56.67 | $1.83 \times 10^5$ | $3.92 \times 10^4$ | $2.42 \times 10^7$ |
| Handwriting | 64.24 | 0.57 | 58.00 | $2.08 \times 10^5$ | $5.42 \times 10^3$ | $4.65 \times 10^6$ | **0.71** | 72.00 | $1.31 \times 10^5$ | $4.75 \times 10^3$ | $5.56 \times 10^6$ |
| EigenWorms | – | **0.55** | 60.16 | $2.11 \times 10^5$ | $4.50 \times 10^5$ | $5.47 \times 10^8$ | 0.50 | 53.91 | $1.42 \times 10^5$ | $6.83 \times 10^5$ | $7.03 \times 10^8$ |
| MotorImagery | – | 0.55 | 60.00 | $1.49 \times 10^5$ | $7.71 \times 10^5$ | $8.38 \times 10^7$ | **0.55** | 61.00 | $1.58 \times 10^5$ | $8.34 \times 10^5$ | $1.91 \times 10^8$ |
| EthanolConcentration | 23.19 | **0.53** | 54.02 | $1.47 \times 10^5$ | $5.90 \times 10^4$ | $4.71 \times 10^7$ | 0.49 | 49.43 | $2.09 \times 10^5$ | $4.55 \times 10^4$ | $6.47 \times 10^7$ |
| AtrialFibrillation | 20.00 | 0.53 | 53.33 | $1.47 \times 10^5$ | $2.15 \times 10^4$ | $1.76 \times 10^7$ | **0.59** | 60.00 | $2.03 \times 10^5$ | $1.68 \times 10^4$ | $3.14 \times 10^7$ |
| StandWalkJump | 40.00 | 0.49 | 50.00 | $2.01 \times 10^5$ | $9.37 \times 10^4$ | $5.98 \times 10^7$ | **0.66** | 66.67 | $2.10 \times 10^5$ | $8.62 \times 10^4$ | $1.44 \times 10^8$ |
| DuckDuckGeese | – | **0.39** | 46.00 | $1.74 \times 10^5$ | $1.45 \times 10^6$ | $3.40 \times 10^7$ | 0.34 | 40.00 | $1.44 \times 10^5$ | $1.45 \times 10^6$ | $1.44 \times 10^7$ |
| PhonemeSpectra | – | **0.21** | 21.42 | $2.16 \times 10^5$ | $1.52 \times 10^4$ | $8.70 \times 10^6$ | 0.19 | 19.49 | $1.78 \times 10^5$ | $1.24 \times 10^4$ | $1.02 \times 10^7$ |

## 7.3  ADDITIONAL RESULTS FOR MCUNET AND PROXYLESSNAS

We present a comparison between five candidates from the MCUNet model list (Lin et al., 2020; 2021), which are optimized for ImageNet and which we retrained for CIFAR10, a version of MCUNet that we optimized specifically for CIFAR10 using ProxylessNAS (Cai et al., 2018), and our results from Fig. 1, see Table 4. The original input resolution of the CIFAR10 dataset is 32x32 which is used by our models. For the MCUNet models we report different resolutions because 48x48 is the smallest input resolution that MCUNet supported without crashing the ProxylessNAS framework and all other input resolutions were predetermined by the MCUNet architectures. A comparison with the other time series datasets we present in our evaluation is not easily possible, since both MCUNet and ProxylessNAS do not support input with only one spatial dimension. Since our tool reports FLOPs instead of MACs, we assume that one MAC operation (multiply-accumulate) equals two FLOPs, i.e., a multiplication followed by an addition. Since MCUNet uses a different mapping tool than we do, we also report the SRAM and Flash consumption of the mapping achieved by our tool in parentheses.

In general, we observe that our NAS approach was able to find network candidates that achieve a slightly higher accuracy than all tested MCUNet variants, while offering an overall better memory footprint, especially in terms of ROM consumption, at the cost of being more computationally complex. One reason for the higher computational complexity may be that MCUNet uses computationally less intensive depthwise separable convolutions (Chollet, 2017), while our optimized architecture, i.e. ResNet18, uses regular convolutions. In addition, we noticed that the dynamic first-fit memory allocator of the DNN mapping tool we used often reported significantly lower RAM consumption for smaller input resolutions than MCUNet's patch-based strategy, while the opposite was true for larger input sizes. We also report results for a version of MCUNet for CIFAR10 (mcunet-proxylessnas) which we explicitly searched for using ProxylessNAS using the default configuration provided by the authors and optimizing for 300 steps using CIFAR10 for training and validation. Compared to our approach which provides a Pareto-front of trained and ready to deploy DNN models, ProxylessNAS emits a single MCUNet configuration deemed optimal and estimations about the configurations's

Table 4: Comparative results of MCUNet and our approach for CIFAR10. MCUNet-in0-4 are taken from the MCUNet model list, while mcunet-proxyless was optimized specifically for CIFAR10 using ProxylessNAS.

| Model | MACs | SRAM/RAM | Flash/ROM | Top1 (int8) | Input |
|---|---|---|---|---|---|
| mcunet-in0 | 6.4M | 266KB (60KB) | 889KB (573KB) | 80.79% | 48x48px |
| mcunet-in1 | 12.8M | 307KB (147KB) | 992KB (587KB) | 82.55% | 96x96px |
| mcunet-in2 | 67.3M | 242KB (410KB) | 878KB (586KB) | 86.24% | 160x160px |
| mcunet-in3 | 81.8M | 293KB (495KB) | 897KB (594KB) | 86.13% | 176x176px |
| mcunet-in4 | 125.9M | 456KB (614KB) | 1876KB (1437KB) | 87.60% | 160x160px |
| mcunet-proxyless | 25.5M | 971KB (83KB) | 2923KB (2842KB) | 82.33% | 48x48px |
| ours-small | 37M | 21KB | 43KB | 81.00% | 32x32px |
| ours-medium | 54M | 24KB | 106KB | 85.00% | 32x32px |
| ours-large | 81.9M | 276KB | 212KB | 87.96% | 32x32px |

accuracy and FLOPS ($86.53\%$ and $27.16M$ estimated) which we could then used separately for training. The actual performance of the trained DNN turned out relatively accurate although slightly overestimated in comparison to the estimated values ($82.33\%$ and $25.5M$ actual).

## 7.4 SYNTHETIC MINIMIZATION PROBLEM

In Fig. 5 we provide the topography of the synthteic minimization problem we consider in Section 5.4. The global minima can be found at $\theta_0 = 15$, $\theta_1 = 5$ and the problem was taken from Passino (2005).

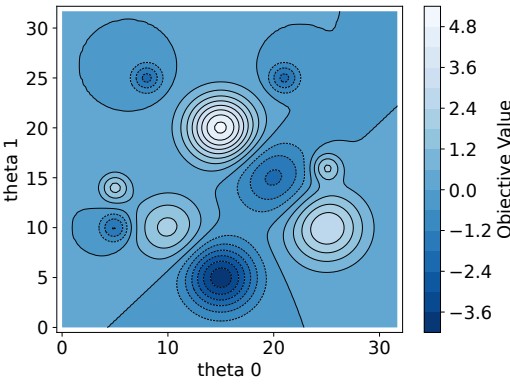

Figure 5: Topography of the synthetic minimization problem.

