# OpenReview forum: "Neural Architecture Search for TinyML with Reinforcement Learning"
_ICLR.cc/2024/Conference — ICLR 2024 Conference Withdrawn Submission_

### Official Review · Reviewer_KEWJ · 2023-10-26

**Soundness:** 2 fair
**Presentation:** 2 fair
**Contribution:** 2 fair
**Rating:** 5
**Confidence:** 4

**Summary:**

This paper introduces a new NAS strategy based on multi-objective Bayesian optimization and an ensemble of competing parametric ARS RL agents. Experiments are conducted on two TinyML use cases, including image classification and time-series classification.

**Strengths:**

1. Multi-objective optimization is critical for improving the performance/efficiency of deep learning on microcontrollers.
2. The studied problem has many practical applications.

**Weaknesses:**

1. The technical contribution of the proposed method is a bit limited. MOBOpt and ARS RL are existing techniques. Combining them and applying them to TinyML is a bit straightforward.

2. This paper lacks experiments on more challenging tasks, such as ImageNet classification on MCU [1] and Tiny Object Detection [2].

3. In addition, this paper also lacks direct comparisons with stronger baselines [1,2].

[1] Lin, Ji, et al. "Mcunet: Tiny deep learning on iot devices." Advances in Neural Information Processing Systems 33 (2020): 11711-11722.

[2] Lin, Ji, et al. "Memory-efficient patch-based inference for tiny deep learning." Advances in Neural Information Processing Systems 34 (2021): 2346-2358.

**Questions:**

See comments above

---

> ### Author Response · Authors · 2023-11-14
> **Response**
>
> We thank the reviewer for his time and effort in reviewing our paper and are pleased to hear that the reviewer considers our approach to be practical.
>
> > 1. The technical contribution of the proposed method is a bit limited. MOBOpt and ARS RL are existing techniques. Combining them and applying them to TinyML is a bit straightforward.
>
> We agree with the reviewer that both MOBOpt and ARS are existing techniques that are studied for years. However, we want to highlight that a combination of them (and how we did it) is anything but straightforward. We believe that the novelty lies in how we integrated both techniques and how they both play their strengths without being computationally infeasible.
>
> > 2. This paper lacks experiments on more challenging tasks, such as ImageNet classification on MCU [1] and Tiny Object Detection [2].
>
> We believe that the main application of edge computing is to perform data analysis close to where data is generated, i.e., on sensor nodes. As a result, we argue that most practical use cases will focus on time series analysis rather than image-based classification, although this is not excluded. Therefore, we focus our analysis on time series classification tasks using the UCA&UCR time series analysis benchmark. To keep the tasks challenging, we only consider multivariate problems. To demonstrate the flexibility of our approach, we also applied it to an image-based task, i.e., CIFAR10, see Figure 1.
>
> > 3. In addition, this paper also lacks direct comparisons with stronger baselines [1,2].
>
> We have added comparative results of our approach to MCUNet. As this was requested by several reviewers, you can find it in the general rebuttal above. We hope that these additional results are of interest.
>
> We hope we managed to address all your concerns and are happy to discuss further.

---

### Official Review · Reviewer_CC3e · 2023-10-31

**Soundness:** 2 fair
**Presentation:** 3 good
**Contribution:** 3 good
**Rating:** 5
**Confidence:** 3

**Summary:**

To optimize DNN for deployment on microcontrollers, the paper proposes a NAS approach that combines Augmented Random Search with Reinforcement learning for Bayesian Optimization. It employs an ensemble of competing polices to identify optimal DNN architectures and also strike a balance among accuracy, memory and computational complexity. Experimental results demonstrate the proposed approach outperforms traditional multi-objective Bayesian optimization methods across various datasets and architectures.

**Strengths:**

1. The idea of combining ARS and RL for Bayesian Optimization is novel and interesting.
2. The paper is well-written and clearly presented.

**Weaknesses:**

1. The superiority of the proposed method is not convincing based on the existing experiments. There lacks comparison with NAS methods.
2. It is hard to duplicate the results based on the details provided in the paper.

**Questions:**

1. As a NAS strategy for microcontrollers, why is the proposed method only compared to Bayesian Optimization methods instead of related NAS methods (especially those for microcontrollers)? Also, more recent methods should be compared with instead of those quite traditional ones.
2. Why choose only ParEGO (proposed in 2006) for comparison in Table 1? Also, it seems that the superiority of the proposed method is not quite impressive compared to ParEGO.

---

> ### Author Response · Authors · 2023-11-14
> **Response**
>
> We thank the reviewer for working through our paper and are happy to hear that the reviewer found it interesting.
>
> > The superiority of the proposed method is not convincing based on the existing experiments. There lacks comparison with NAS methods.
>
> > As a NAS strategy for microcontrollers, why is the proposed method only compared to Bayesian Optimization methods instead of related NAS methods (especially those for microcontrollers)? Also, more recent methods should be compared with instead of those quite traditional ones.
>
> We agree with the reviewer and share the concern. As this was requested by multiple reviewers, we added results for MCUNet and ProxylessNAS, a zero-shot hardware aware NAS strategy, please see the general rebuttal above. We hope that these additional results are of interest.
>
> > It is hard to duplicate the results based on the details provided in the paper.
>
> We are checking whether it is possible to publish source code and think that we will come to a solution. Furthermore, we evaluate the possibility to let others reproduce our algorithms and experiments.
>
> > Why choose only ParEGO (proposed in 2006) for comparison in Table 1? Also, it seems that the superiority of the proposed method is not quite impressive compared to ParEGO.
>
> We chose ParEGO as the main comparison in Table 1 because it is the closest competitor to our approach in the experiments we present in Figure 1. Unfortunately, we were not able to obtain additional results for MCUNet, as it only supports image-based datasets and not datasets with only a single spatial dimension.
>
> We hope we managed to address all your concerns and are happy to discuss further.

---

> > ### Comment · Reviewer_CC3e · 2023-11-23
> >
> > Thanks for the clarification and experiments. I have read the author's responses and insist my initial rating.

---

### Official Review · Reviewer_FN6N · 2023-10-31

**Soundness:** 3 good
**Presentation:** 3 good
**Contribution:** 2 fair
**Rating:** 3
**Confidence:** 4

**Summary:**

This paper uses reinforcement learning to perform multi-objective hyper parameter optimization for pruning a given architecture. The hyperparameters identified by the search algorithm are then used by the pruning pipeline to prune and then evaluate the model on the various objectives specified.
 The Augmented Random Search (ARS) agents are trained to identify the next most promising candidates. They circumvent the computationally expensive step of training the candidates to obtain the reward model by employing a gaussian process as a surrogate model to predict the reward. The GP prior is initialized by evaluating candidates sampled using Latin-Hypercube sampling. During every iteration of the search, multiple ARS agents are trained and the GP's posterior is updated by evaluating the best candidates proposed by all the agents. The various objectives used are accuracy, RAM and ROM memory consumption and FLOPS.

  They performed pruning on MobileNetV3 trained on DaLiac , ResNet trained on Cifar10. They also evaluated their algorithm on timeseries classification where the architecture used is CNN. In addition to this, on a synthetic dataset, they also showed that their algorithm is able to find a good minima when compared to ParEgo even during poor initialization where all the samples are in a region with no gradients.

**Strengths:**

1. It outperforms all the ParEgo multi-objective bayesian optimization baseline.
2. ARS agent in conjunction with GP as a surrogate avoids expensive evaluations to obtain the reward model. It would be good to highlight this and report the time taken by all the algorithms.

**Weaknesses:**

1. In the past, other multi-objective algorithms have used Knowledge distillation (KD) rather than pruning to obtain smaller well performing architectures. Please adapt yours to perform KD and compare against the following baselines [5], [6], [7]. It is also essential to indicate the drop in accuracy of the searched architecture when compared to the original architecture. Given that these algorithms also employ reinforcement learning and Bayesian optimization, please highlight how your algorithm is different from theirs and also demonstrate that yours performs better than these baselines.

2. Please cite other multi-objective NAS algorithms such as  MnasNet [1], once-for-all [2], NSGA-NET [3], Dppnet [4] in the related work.
3. Given that your algorithm is performing HPO, please include other HPO solvers that use multi-objective algorithms such as NSGA-II, Multi-Objective Particle Swarm Optimization etc.

[1] Once-for-All: Train One Network and Specialize it for Efficient Deployment, Cai et al.
[2] MnasNet: Platform-Aware Neural Architecture Search for Mobile, Tan et al.
[3] NSGA-Net: Neural Architecture Search using Multi-Objective Genetic Algorithm, Lu et al.
[4] DPP-Net: Device-aware Progressive Search for Pareto-optimal Neural Architectures, Dong et al.
[5] Learnable embedding space for efficient neural architecture compression., Can et al.
[6] AMC: automl for model compression and acceleration on mobile devices., He et al.
[7] N2N learning: Network to network compression via policy gradient reinforcement learning., Ashok et al.

**Questions:**

1. How can a weighted sum reward yield various Pareto fronts? Given that the final objective is a weighted sum, one must vary the weights to obtain various fronts. This is not a Pareto-optimal solution. Algorithms such as NSGA-II yield pareto-optimal solution.
2. How can k-means ensure the diversity of the solutions in Pareto-front? K-means selects solutions from each centroid which would imply that it is the most representative sample in that cluster. To ensure diversity, we want to sample solutions from region where the number of solutions in the neighborhood is sparse.  NSGA-II uses crowding distance, Lemonade uses kernel density estimator and in both cases they select solutions from less crowded region.

---

> ### Author Response · Authors · 2023-11-14
> **Response**
>
> We thank the reviewer for taking the time and effort to thoroughly review our paper. We are glad that the effectiveness of our approach is recognized. As with all black-box NAS techniques, the biggest influence on the optimization time is the time needed to train the DNNs. Since we use small, inexpensive RL agents, their additional overhead has proven to be negligible compared to other black-box solvers.
>
> > In the past, other multi-objective algorithms have used Knowledge distillation (KD) rather than pruning to obtain smaller well performing architectures. Please adapt yours to perform KD and compare against the following baselines [5], [6], [7]. It is also essential to indicate the drop in accuracy of the searched architecture when compared to the original architecture. Given that these algorithms also employ reinforcement learning and Bayesian optimization, please highlight how your algorithm is different from theirs and also demonstrate that yours performs better than these baselines.
>
> > Please cite other multi-objective NAS algorithms such as MnasNet [1], once-for-all [2], NSGA-NET [3], Dppnet [4] in the related work.
>
> We thank the reviewer for highlighting important additional related work which we indeed did not properly include. We made sure to update the related work section of our work accordingly (see updated paper). We also extended our related work section to reflect advances in the usage of knowledge distillation techniques to achieve model compression besides other well-known compression techniques like pruning and quantization.
>
> > Given that your algorithm is performing HPO, please include other HPO solvers that use multi-objective algorithms such as NSGA-II, Multi-Objective Particle Swarm Optimization etc.
>
> We provide comparative results for two datasets and model architectures using NSGA-II in Figure 1. In general, the reason we decided to focus on Bayesian optimization rather than evolutionary or swarm-based strategies is that they tend to have a high sampling complexity, making them underperform in scenarios where very little sampling can be done, we which believe can also be seen in the results provided in Fig. 1.
>
> > How can a weighted sum reward yield various Pareto fronts? Given that the final objective is a weighted sum, one must vary the weights to obtain various fronts. This is not a Pareto-optimal solution. Algorithms such as NSGA-II yield pareto-optimal solution.
>
> We perform multi-objective optimization, i.e., we have multiple objective metrics, 4 in your case. As a result, the result of the optimization is a 4-dimensional Pareto front. Since this front cannot be visualized directly in 2D or 3D space, we decided to use a sliced representation, i.e., all-versus-accuracy in Figure 1. More precisely, the Pareto fronts provided in Figure 1 are the same front for each row, just from three different "perspectives". To provide a single reward signal to the RL agents, we compute a weighted sum of the 4D target vector. The reward is not visualized in the Pareto fronts. The weights for the sum can either be provided by the user based on the subjective importance of the goals, or the algorithm randomly chooses them for each trial to provide a diverse set of tradeoffs.
>
> > How can k-means ensure the diversity of the solutions in Pareto-front? K-means selects solutions from each centroid which would imply that it is the most representative sample in that cluster. To ensure diversity, we want to sample solutions from region where the number of solutions in the neighborhood is sparse. NSGA-II uses crowding distance, Lemonade uses kernel density estimator and in both cases they select solutions from less crowded region.
>
> We use K-means to find representative samples that are spread far apart as good starting points for our RL agents to search for the next candidate during optimization. Since we are using multiple local agents searching in parallel, we wanted to make sure that they are distributed throughout the search space to ensure diversity in their proposed solutions.
>
>
> We like to thank the reviewer again for his excellent points in helping us clarify the paper's contents. We would be happy to discuss further in case additional questions arise.

---

### Official Review · Reviewer_4rfG · 2023-10-31

**Soundness:** 3 good
**Presentation:** 2 fair
**Contribution:** 2 fair
**Rating:** 5
**Confidence:** 4

**Summary:**

This study merges the efficiency of Gaussian Processes (GPs) with the effectiveness of Reinforcement Learning (RL) agents to facilitate the efficient exploration of Deep Neural Network (DNN) architectures that can be directly implemented on microcontrollers. Unlike previous research on RL for Multi-Objective Optimization (MOOpt), it doesn't apply RL directly to the MOOpt problem's search space. Instead, it employs RL to enhance the optimization process conducted on GPs. To tackle the issue of resource-intensive RL agent training, it suggests training a group of competing policies within MOBOpt. These policies utilize a straightforward Augmented Random Search (ARS) method for candidate sampling.

**Strengths:**

1. The research introduces an innovative approach by uniting the sampling efficiency of Gaussian Processes (GPs) with the expressive capabilities of Reinforcement Learning (RL) agents. Additionally, the incorporation of Augmented Random Search represents a novel enhancement.

2. The study demonstrates performance improvements when compared to alternative Bayesian optimization strategies.

**Weaknesses:**

1. From a technical standpoint, I haven't identified a direct link between the proposed method and TinyML, and this study hasn't delved deeply into the realm of TinyML models, and some important related works [1] [2]  are missing.

2. The comparative methods in this study are restricted to Bayesian optimization strategies, omitting comparisons with other Neural Architecture Search (NAS) approaches such as zero-shot methods, optimization-based methods, or predictor-based methods.

3. The mentioned validated architectures are limited to ResNet18 and MobileNetV3, which are somewhat dated in the context of contemporary developments.

[1] MCUNet: Tiny Deep Learning on IoT Devices
[2] MCUNetV2: Memory-Efficient Patch-based Inference for Tiny Deep Learning

**Questions:**

1. The main performance comparison didn't conclude for some architectural performance but mainly focused on the comparison with other optimization methods. How about the performance comparison with MCUNetV1[1], and MCUNetV2 [2]?
2. What is the search cost and efficiency compared with other types of NAS methods? For example, it would be fair to compare with other zero-shot NAS methods or preditor-based methods？

---

> ### Author Response · Authors · 2023-11-14
> **Response**
>
> First and foremost we want to thank the reviewer for the time and effort spent on reviewing our paper. We are happy to hear that our approach is considered innovative and novel.
>
> > From a technical standpoint, I haven't identified a direct link between the proposed method and TinyML, and this study hasn't delved deeply into the realm of TinyML models, and some important related works [1] [2] are missing.
>
> We thank the reviewer for highlighting important related work and have updated our work accordingly (see updated paper). The challenge of architecture design for TinyML lies in finding a trade-off between hardware constraints of a given edge system and DNN accuracy. We tried to automate the hardware aware architectural search for a given target platforms, represented by a set of constraints, e.g., <256 Kb RAM, <1MB Flash, on the target metrics, i.e., ROM, RAM, FLOPs, accuracy, by combing multi-objective NAS (black-box Bayesian optimization) with RL and comparing it to other optimization strategies.
>
> > The comparative methods in this study are restricted to Bayesian optimization strategies, omitting comparisons with other Neural Architecture Search (NAS) approaches such as zero-shot methods, optimization-based methods, or predictor-based methods.
>
> Thank you for bringing up that point! As also requested by other reviewers, we now provide more results for MCUNet and ProxylessNAS, a zero-shot optimization framework, (see general rebuttal response above). We think that these additional results and especially the comparison with ProxylessNAS are of interest to the reviewers and readers of the paper.
>
> > The mentioned validated architectures are limited to ResNet18 and MobileNetV3, which are somewhat dated in the context of contemporary developments.
>
> We selected ResNet18 and MobileNetV3 as they are architectures commonly used in TinyML benchmarks like MLPerf Tiny [1]. Furthermore, these architectures cover all relevant layer types and network structures that are also used in other TinyML architectures such as MCUNet.
>
> [1] Banbury, Colby, et al. "MLPerf Tiny Benchmark." Thirty-fifth Conference on Neural Information Processing Systems Datasets and Benchmarks Track (Round 1). 2021.
>
> > The main performance comparison didn't conclude for some architectural performance but mainly focused on the comparison with other optimization methods. How about the performance comparison with MCUNetV1[1], and MCUNetV2 [2]?
>
> > What is the search cost and efficiency compared with other types of NAS methods? For example, it would be fair to compare with other zero-shot NAS methods or preditor-based methods?
>
> As with all black-box optimization approaches, the main factor influencing the search cost of our approach is the time required to train DNN architecture candidates during optimization. We found that the additional overhead of the RL agents we propose to use in our work is negligible compared to other black-box solvers, e.g., ParEGO.
>
> We agree that the comparison between slower but accurate black-box optimization and faster but imprecise zero-shot NAS methods is interesting. As a result, we provide additional results for MCUNet and ProxylessNAS for CIFAR10 and compare them with the results we report in our work (see general rebuttal response).
>
> We hope we managed to address all your concerns and are happy to discuss further.

---

### Official Review · Reviewer_obe6 · 2023-11-01

**Soundness:** 3 good
**Presentation:** 3 good
**Contribution:** 2 fair
**Rating:** 5
**Confidence:** 2

**Summary:**

This work proposes a NAS strategy specifically for TinyML using multi-objective Bayesian optimization (MOBOpt) and an ensemble of competing parametric policies trained using Augmented Random Search (ARS) Reinforcement Learning (RL) agents. This approach can be utilized to explore the design tradeoffs between a DNN’s predictive accuracy, memory consumption on a given target system, and computational complexity. The experiments show competitive results compared to prior optimization strategies. The paper is well organized in general, but my major concern is that the proposed optimization approaches such as MOBOpt are generic and not specific to TinyML. Particularly, the authors argue that it is expensive to evaluate the accuracy and memory consumption for TinyML system, but accuracy evaluation is required for NAS despite the targeted computing platform. Although memory (RAM and ROM) consumption is a specific requirement for TinyML system, the evaluation is similar or even less expensive than accuracy evaluation. Hence, more discussion about why tinyML makes NAS challenging and necessary is expected.

**Strengths:**

The proposed NAS approach produces competitive models for MCU platform. The paper is well organized.

**Weaknesses:**

The proposed NAS optimization seems to be generic and not specific for TinyML platform, which makes the research problem not quite convincing.

**Questions:**

Could you illustrate why tinyML makes NAS challenging and unqiue?
How the proposed NAS optimization applies specifically for tinyML rather than general network design?

---

> ### Author Response · Authors · 2023-11-14
> **Response**
>
> We thank the reviewer for the time and effort working through our paper. We are happy to hear that our approach is competitive and meaningful. We also now provide more results compared to MCUnet, which has been suggested by some reviewers (see the general response).
>
> The main challenge in deploying DNNs on the edge, i.e., tinyML, are the strict resource constraints, e.g., <256 Kb RAM and <1Mb Flash, in comparison to the high computational complexity and memory demands of most used DNN architectures. This makes finding DNN architectures that can achieve sufficient accuracy on relevant ML problems while being small enough to be feasibly deployable on edge systems a complicated and hardware-dependent process. NAS provides a great way to search for DNN architectures, but usually only focuses on network accuracy and not so deeply on the tradeoff between hardware constraints and accuracy. Furthermore, many existing NAS techniques that do consider the target hardware focus on zero-shot techniques that rely on fast but often imprecise predictive models instead of training DNN candidates. As a result, our work aims to explore the feasibility and explore the effectiveness of considering hardware aware NAS as a multi-objective optimization process in combination with reinforcement learning.
>
> We hope that we have been able to explain sufficiently why tinyml places special demands on the optimization of neural networks. If there are any remaining or additional questions we are happy to further discuss them with you.

---

> > ### Comment · Reviewer_obe6 · 2023-11-22
> >
> > Thanks for the clarification and it is good to see the comparison with MCUNet.
> > I agree with other reviewers that the major contribution is a combination of existing techniques, which makes the novelty and contribution insufficient. So I insist my prior rating.

---

### Author Response · Authors · 2023-11-14
**General Rebuttal**

We would like to thank the reviewers for their time and effort they have spent on our paper. The comments and suggestion helped us to improve our work.

As requested by some reviewers, we present a comparison between five candidates from the MCUNet model list [1], which are optimized for ImageNet and which we retrained for CIFAR10, a version of MCUNet that we optimized specifically for CIFAR10 using ProxylessNAS [2], and our results, see Fig. 1. A comparison with the other time series datasets we present in our evaluation is not easily possible, since both MCUNet and ProxylessNAS do not support input with only one spatial dimension. The original input resolution of the CIFAR10 dataset is 32x32 which is used by our models. For the MCUNet models we report different resolutions because 48x48 is the smallest input resolution that MCUNet supported without crashing the ProxylessNAS framework and all other input resolutions were predetermined by the MCUNet architectures.  Since our tool reports FLOPs instead of MACs, we assume that one MAC operation (multiply-accumulate) equals two FLOPs, i.e., a multiplication followed by an addition. As MCUNet uses a different mapping tool than we do, we also report the SRAM and Flash consumption of the mapping achieved by our tool in parentheses additionally to the results reported by MCUNet.

| Model               | MACs                     | SRAM/RAM       | Flash/ROM         | Top1 (int8)               | Input Resolution |
|---------------------|--------------------------|----------------|-------------------|---------------------------|------------------|
| mcunet-in0          | 6.4M                     | 266KB (60KB)   | 889KB (573KB)     | 80.79%                    | 48x48px          |
| mcunet-in1          | 12.8M                    | 307KB (147KB)  | 992KB (587KB)     | 82.55%                    | 96x96px          |
| mcunet-in2          | 67.3M                    | 242KB (410KB)  | 878KB (586KB)     | 86.24%                    | 160x160px        |
| mcunet-in3          | 81.8M                    | 293KB (495KB)  | 897KB (594KB)     | 86.13%                    | 176x176px        |
| mcunet-in4          | 125.9M                   | 456KB (614KB)  | 1876KB (1437KB)   | 87.60%                    | 160x160px        |
| mcunet-proxylessnas | 25.5M (27.16M predicted) | 971KB (83KB)   | 2923KB (2842KB)   | 82.33% (86.53% predicted) | 48x48px          |
| ours-small          | 37M                      | 21KB           | 43KB              | 81.00%                    | 32x32px          |
| ours-medium         | 54M                      | 24KB           | 106KB             | 85.00%                    | 32x32px          |
| ours-large          | 81.9M                    | 276KB          | 212KB             | 87.96%                    | 32x32px          |

In general, we observe that our NAS approach was able to find network candidates that achieve a slightly higher accuracy than all tested MCUNet variants, while offering an overall better memory footprint, especially in terms of ROM consumption, at the cost of being more computationally complex. One reason for the higher computational complexity may be that MCUNet uses computationally less intensive depth-wise separable convolutions [3], while our optimized architecture, i.e. ResNet18, uses regular convolutions. In addition, we noticed that the dynamic first-fit memory allocator of the DNN mapping tool we used often reported significantly lower RAM consumption for smaller input resolutions than MCUNet's patch-based strategy, while the opposite was true for larger input sizes.

We also report results for a version of MCUNet for CIFAR10 (mcunet-proxylessnas) which we explicitly searched for using ProxylessNAS [2] using the default configuration provided by the authors and optimizing for 300 steps using CIFAR10 for training and validation. Compared to our approach which provides a Pareto-front of trained and ready to deploy DNN models, ProxylessNAS emits a single MCUNet configuration deemed optimal and estimations about the configurations's accuracy and FLOPS (86.53% and 27.16M estimated) which we could then use separately for training. The actual performance of the trained DNN turned out relatively accurate although slightly overestimated in comparison to the estimated values (82.33% and 25.5M actual).


[1] Lin, Ji, et al. "Mcunet: Tiny deep learning on iot devices." Advances in Neural Information Processing Systems 33 (2020): 11711-11722.

[2] Cai, Han, Ligeng Zhu, and Song Han. "ProxylessNAS: Direct Neural Architecture Search on Target Task and Hardware." International Conference on Learning Representations. 2018.

[3] Chollet, François. "Xception: Deep learning with depthwise separable convolutions." Proceedings of the IEEE conference on computer vision and pattern recognition. 2017.

---

### Comment · Area_Chair_65qJ · 2023-11-22
**Author-Reviewer Discussions**

Dear reviewers,

The authors have provided detailed responses to your comments. Please read through them carefully to see whether your concerns have been addressed.

Best, AC